# In Silico Design of miniACE2 Decoys with In Vitro Enhanced Neutralization Activity against SARS-CoV-2, Encompassing *Omicron Subvariants*

**DOI:** 10.3390/ijms251910802

**Published:** 2024-10-08

**Authors:** Jenny Andrea Arévalo-Romero, Gina López-Cantillo, Sara Moreno-Jiménez, Íñigo Marcos-Alcalde, David Ros-Pardo, Bernardo Armando Camacho, Paulino Gómez-Puertas, Cesar A. Ramírez-Segura

**Affiliations:** 1Unidad de Ingeniería Celular y Molecular, Instituto Distrital de Ciencia, Biotecnología e Innovación en Salud, IDCBIS, Bogotá 111611, Colombia; jandrea.arevalo@gmail.com (J.A.A.-R.); glopez@idcbis.org.co (G.L.-C.); svmoreno@idcbis.org.co (S.M.-J.); bacamacho@idcbis.org.co (B.A.C.); 2Instituto de Errores Innatos del Metabolismo, Facultad de Ciencias, Pontificia Universidad Javeriana, Bogotá 110231, Colombia; 3Grupo de Modelado Molecular del Centro de Biología Molecular Severo Ochoa, 14 CSIC-UAM, 28049 Madrid, Spain; imarcos@cbm.csic.es (Í.M.-A.); davidrp@cbm.csic.es (D.R.-P.)

**Keywords:** ACE2-decoys, miniACE2, SARS-CoV-2 variants evading monoclonal antibodies, next-generation treatments for SARS-CoV-2, in silico design of next-generation therapies for infectious diseases, basic science for translational medicine

## Abstract

The COVID-19 pandemic has overwhelmed healthcare systems and triggered global economic downturns. While vaccines have reduced the lethality rate of SARS-CoV-2 to 0.9% as of October 2024, the continuous evolution of variants remains a significant public health challenge. Next-generation medical therapies offer hope in addressing this threat, especially for immunocompromised individuals who experience prolonged infections and severe illnesses, contributing to viral evolution. These cases increase the risk of new variants emerging. This study explores miniACE2 decoys as a novel strategy to counteract SARS-CoV-2 variants. Using in silico design and molecular dynamics, blocking proteins (BPs) were developed with stronger binding affinity for the receptor-binding domain of multiple variants than naturally soluble human ACE2. The BPs were expressed in *E. coli* and tested in vitro, showing promising neutralizing effects. Notably, miniACE2 BP9 exhibited an average IC_50_ of 4.9 µg/mL across several variants, including the Wuhan strain, Mu, Omicron BA.1, and BA.2 This low IC50 demonstrates the potent neutralizing ability of BP9, indicating its efficacy at low concentrations.Based on these findings, BP9 has emerged as a promising therapeutic candidate for combating SARS-CoV-2 and its evolving variants, thereby positioning it as a potential emergency biopharmaceutical.

## 1. Introduction

SARS-CoV-2, the causative agent of the COVID-19 pandemic, emerged in Wuhan in December 2019 [1,2,3,4] and has since been associated with 776,007,137 confirmed cases and 7,059,612 fatalities as of September 2024 [5]. Characterizing the virus’s molecular structure was essential for developing biotechnological interventions that helped mitigate the pandemic and facilitate a return to normalcy. However, concerns about the rapid evolution of the virus driven by genetic mutations, viral recombination [6], and selective immune pressure [7,8] have led to a decrease in the efficacy of prophylactic measures, such as vaccines and monoclonal antibodies, addressing new virus mutations [9], particularly in emergency cases [10]. Although cell memory mediated by T cells confers cross-protection against new Omicron subvariants [11], the emergence of mutations associated with L455S and F456L in the current circulating variants BA.2.86, JN.1, KP2, KP3, and LB.1, keeps raising new alarms after exhibiting extensive immune evasion [12,13], and is associated with a potential infection of lung cells through in vitro approximations [14].

The data presented in this manuscript underscore the importance of developing broad-spectrum therapeutic strategies against SARS-CoV-2 variants, which are currently being characterized in innovative projects [15,16,17]. Current strategies are effective against a limited number of variants, requiring repeated resource investment to maintain adequate protection [18], particularly for at-risk populations, such as immunosuppressed individuals [19]. Furthermore, the limited applicability of monoclonal antibodies as an emergency tool against SARS-CoV-2 due to the virus’s high mutation rate highlights the need for ongoing development and testing of biological models for emergency use. As the coronavirus continues to adapt, the potential for future outbreaks remains, underscoring the urgency of these efforts.

ACE2 decoys have emerged as promising therapeutic alternatives in recent years [20], inspired by the pathophysiological interaction between SARS-CoV-2 and host cells. This interaction involves binding of the viral spike protein receptor-binding domain (RBD) to the human angiotensin-converting enzyme 2 (hACE2) receptor on host cells. In this study, we utilize the binding interface of hACE2, which directly interacts with the RBD, as a template to design miniature versions of hACE2. These engineered decoy proteins act as neutralizing agents to counteract SARS-CoV-2 infections.

The application of ACE2 decoys as a therapeutic strategy offers a promising approach to prevent the entry of *Sarbecoviruses* into host cells. These decoy molecules, designed to mimic the ACE2 receptor, effectively sequester the virus by binding to it, thereby obstructing its ability to interact with genuine hACE2 receptors on the cell surface. This mechanism provides a broad-spectrum antiviral solution, targeting the fundamental entry pathway utilized by various *Sarbecovirus* agents. This innovative application of ACE2 decoys stands to be significantly enhanced through translational medicine approaches, such as nebulization, which allows for direct and efficient delivery to the respiratory tract, potentially increasing therapeutic efficacy. Additionally, ACE2 decoys offer promising avenues for addressing persistent viral activity, a key factor in the development of long-term COVID-19, particularly among immunocompromised individuals [21,22,23].

Consequently, ACE2 decoys represent a versatile and potentially effective intervention for managing infections caused by multiple viral variants through a single unified approach [24,25]. While preliminary results are encouraging, further characterization and rigorous clinical studies are essential to fully unlock their therapeutic potential and optimize their application in acute and chronic SARS-CoV-2 infections. In this context, the development and testing of our miniACE2 decoy models—specifically BP9 and BP11, derived from in silico assays—demonstrate potential efficacy against SARS-CoV-2 Omicron subvariants. Given the high mutation rate of the Omicron variant, these miniACE2 decoys could play a crucial role in counteracting its adaptive evolution, underscoring their significance in ongoing efforts to combat COVID-19.

## 2. Results

### 2.1. In Silico Design, Assessment, and Selection of miniACE2 Decoy Proteins

Through analysis of the RBD/hACE2 interaction, based on the PDB: 6M0J structure [26], key regions of ACE2, specifically amino acids S19–S106 and Q340–A386, corresponding to the interaction surface, were identified for incorporation into the miniACE2 design. These discontinuous conformational regions of the ACE2 protein were linked in various ways to reconstruct the RBD-interacting surface, while excluding both the catalytic domain and non-interacting regions of ACE2.

The sequences of the fragments joined were used to model the BPs 3D structures using the Phyre2 [27] and Robetta servers [28]. The 3D models obtained were visualized using the PyMOL graphical environment [29]. All models were aligned with the coordinates of the PDB entry 6M0J [26], maintaining the RBD structure at its spatial location. The most promising candidates were selected for Molecular Dynamics (MD) simulations by conducting a preliminary visual evaluation and analyzing interactions between structures. These proteins were named BP1, BP2, BP3, BP4, BP5, BP6, BP8, BP9, BP10 and BP11 (Appendix A).

The proteins selected for interaction with the RBD were subjected to 100 ns of unrestricted MD simulation (Appendix A). The files obtained from the MD simulations were evaluated using the PRODIGY web server [30] to obtain the in silico estimated values for ΔG and K_d_. To assess the potential increased capacity to interact with the RBD compared with the natural ACE2 receptor, the ACE2 K_d_/BP K_d_ ratio was calculated for each BP (Table 1), and the top five BPs according to the highest ACE2 K_d_/BP K_d_ ratios were selected (BP11, BP9, BP4, BP2, and BP1) (Table 1). Out of these 5 BPs, those with the most stable structures over the 100 ns MD simulations, as indicated by the lowest RMSD values (Table 1), were selected to continue the in silico study. Consequently, three proteins exhibiting the highest ACE2 K_d_/BP K_d_ ratios and the most stable structures (BP9, BP2, and BP11) were chosen as candidate BPs with lengths ranging from 118 to 132 amino acids (Appendix A, Table 1).

### 2.2. Chosen BPs Exhibit Enhanced Efficacy against Diverse SARS-CoV-2 Variants in In Silico Assays

Given that miniACE2 decoys were engineered using the binding surface of hACE2, which directly interacts with the RBD, it is theoretically possible for miniACE2 decoys to overcome SARS-CoV-2 escape variants. This is because the virus is unlikely to evade miniACE2-mediated neutralization without simultaneously decreasing its affinity for native ACE2, which prevents its entry into target cells [17]. Therefore, we hypothesized that RBD variants should be neutralized effectively, if not more so, by miniACE2 decoys, compared to the natural hACE2 receptor.

To test in silico this hypothesis, we prepared BP2, BP9, and BP11 miniACE2 proteins in interaction with various SARS-CoV-2 variants (Wuhan, PDB: 6m0j [26]; Alpha B.1.1.7, PDB: 8DLK [31]; Beta, PDB: 7VX4 [32]; Delta, PDB: 7W9I [33]; Epsilon, PDB: 8DLV [31]; Gamma, PDB: 8DLQ [31]; Kappa, PDB: 7V86 [34]; Omicron BA.2, PDB: 7ZF7 [35]; Omicron BA.3, PDB: 7XB1 [36]; Omicron BA.1, PDB: 7U0N [37]; Omicron BA.1.1, PDB: 7XAZ [36]; Omicron XBB.1, PDB: 8IOV [38]; Omicron BA.2.75, PDB: 8ASY [39]; Omicron BQ.1.1, PDB: 8IF2 [40]; Omicron BA.4/5, PDB: 8AQS [41]). Structural alignment was performed for each PDB file using the Pymol graphical environment to build the interacting models [29]. The obtained structures were subjected to 200 ns of MD simulation. Post-MD, each model was evaluated using the PRODIGY web server [30] (Figure 1, Table 2 and Table 3).

The obtained results indicated that the three miniACE2 decoys (BP9, BP2, and BP11) were able to interact in silico with all tested SARS-CoV-2 variants with better ΔG and K_d_ values than the natural ACE2 receptor. Although BP2 shows the worst values in interaction with the Omicron BA.3 variant (O_BA.3), it has the lowest interaction values among the miniACE2 decoys, while BP11 exhibits the highest interaction values. Additionally, it is worth highlighting that as the affinity for the hACE2/RBD interaction decreases with the Omicron variants, a similar effect is observed in the ΔG and K_d_ values for the BPs. Interestingly, BP9 demonstrates the highest interaction values among the three Omicron subvariants (O_BA.1, O_XBB.1, and O_BA.4/5).

### 2.3. BPs Proteins Produced in E. coli Expression System Demonstrated Enhanced Protection against SARS-CoV-2

Blocking proteins BP9 and BP11, designed in silico, were produced in *E. coli*, yielding stability at high concentrations. They were tested using the GenScript cPass™ SARS-CoV-2 Neutralization Antibody Detection Kit with RBD for the Wuhan, Mu, Omicron BA.1, and Omicron BA.2 variants. Technical independent triplicates (*n* = 3) confirmed that the efficacy observed in the in silico analyses was reproduced in the ELISA neutralizing experiments. hACE2 was used as a benchmark to compare the effectiveness of decoys. The potential of the BP9 and BP11 decoys was confirmed (Figure 2A), demonstrating their effectiveness as shown by in silico assays.

Further assessment of BP9 and BP11′s potential was conducted by evaluating the IC_50_ values. The IC_50_ values for the miniACE2 decoys ranged between 4.24 µg/mL and 18.19 µg/mL, compared to the control hACE2, which displayed IC_50_ values between 92.7 µg/mL and 120.9 µg/mL. Notably, BP9 demonstrated particularly promising performance in In vitro assays, with IC_50_ values consistently between 4.24 to 5.33 µg/mL across all four variants. Given its broad protective effect, BP9 shows potential as a therapeutic tool against SARS-CoV-2 infection (Figure 2A,B, Table 4). Additionally, the statistical significance showed a strong value for the IC_50_ obtained from the BPs compared to the hACE2 (Figure 2C).

### 2.4. Influence of Temperature and Serum Matrix on Protein Stability

The cPass™ SARS-CoV-2 Neutralization Antibody Detection Kit for the BA.1 virus variant was selected as a model, with the hACE2 protein as a benchmark for assessing the stability of BP9 and BP11 under temperature and serum matrix conditions. The ACE2 IC_50_ value for the BA.1 variant (92.92 µg/mL) was used as a standard protein concentration to evaluate the stability of hACE2, BP9, and BP11 under these conditions. To test stability, the proteins were incubated at room temperature and 37 °C for two hours, both in the presence and absence of human serum samples collected prior to the pandemic, confirmed to lack neutralizing antibodies against any SARS-CoV-2 variant.

Our findings show that the neutralization capacity of both BP9 and BP11 remained stable, while an increase in neutralization capacity was observed for hACE2 at 37 °C. In the serum matrix, a reduction in neutralization capacity was observed for all three proteins, with hACE2 showing the greatest decrease, while BP9 exhibited a less pronounced decline. To assess whether there were statistically significant differences in neutralization capacity between hACE2 and proteins BP9 and BP11, the temperature stability coefficient was calculated by dividing the mean percentage of neutralization at 37 °C by the mean percentage of neutralization at room temperature. Similarly, the stability coefficient in the serum was calculated by dividing the mean percentage of neutralization in the serum by the mean percentage of neutralization for each protein in the absence of serum.

These results indicate that the stability of hACE2 appears to increase after 2 h of incubation at 37 °C, whereas BP9 and BP11 maintain their neutralization capacity at this temperature. In terms of stability in the serum matrix, BP9, followed by BP11, demonstrated greater stability, while hACE2 showed a significant decrease in stability within this environment. Statistical analysis was performed using one-way ANOVA (Table 5, Figure 3).

## 3. Discussion and Conclusions

SARS-CoV-2, the agent responsible for the recent global pandemic, was initially underestimated due to its relatively low mutation rate compared to other high-risk viruses like Influenza. However, a key feature of SARS-CoV-2 is its capacity for interspecies recombination, an adaptive mechanism that enhances its survival and poses a risk of future outbreaks. Its ability to recombine, coupled with its presence in animals closely related to humans, highlights the potential for the virus to re-emerge during subsequent epidemics [42,43,44].

Next-generation treatments for SARS-CoV-2, such as mRNA vaccines, have demonstrated superior efficacy compared to traditional methods [45,46,47]. Nevertheless, innovative approaches like cell therapies and ACE2 decoys are still undergoing clinical trials. Preliminary findings on these novel therapies are encouraging, indicating that they are safe and may benefit individuals at higher risk of severe COVID-19 [48,49,50,51,52].

Pathophysiological studies on SARS-CoV-2 infection have highlighted the potential utility of recombinant hACE2 proteins, demonstrating their effectiveness in neutralizing a wide range of SARS-CoV-2 variants in vitro. Therefore, ACE2 decoys are emerging as promising candidates for emergency treatments [53,54,55]. Here, utilizing the PDB structures 6LZG [56] and 6M0J [26], which were available at the onset of the pandemic, three miniACE2 proteins were designed. BP2 was created to include the RBD-interacting surface, incorporating glycosylated asparagine residues N53 and N90 from the ACE2 sequence (corresponding to N33 and N70 on BP2). These positions were mutated in the BP9 and BP11 miniACE2 proteins to N53Q and N90Q. BP9 and BP11 share the same sequence, with one exception; in BP11, positions E37 and D38 were inverted relative to their positions in ACE2, BP9, and BP2 (Appendix A). Although in silico analyses suggested that the inversion of these amino acids would enhance the interaction between BP11 and the RBD compared to BP9, in vitro assays revealed the opposite effect; the IC_50_ value for BP9 ($\bar{X}$ = 4.90) was lower than that for BP11 ($\bar{X}$ = 17.24), indicating a stronger binding affinity for BP9.

Although no adverse effects are anticipated from the natural activity of ACE2, given that the catalytic site is absent in miniACE2, it remains essential to evaluate the potential immune responses or interactions with other human proteins that could lead to toxicity. This is particularly important since linker amino acid sequences were incorporated into the miniACE2 design, which may introduce unforeseen immunogenicity or off-target effects. Therefore, thorough preclinical testing will be crucial to assess safety.

The miniACE2 designed here represents a promising candidate for next-generation treatments, particularly through translational medicine approaches, such as inhalation. This method offers several advantages over conventional systemic administration, including superior bioavailability, lower risk of systemic toxicity, rapid onset of action, reduced dosing, and the ability to deliver higher local concentrations directly into the respiratory tract [57,58]. For example, Urano et al. [59] demonstrated that an inhaled ACE2 decoy achieved therapeutic efficacy in rodents at a 20-fold lower dose than intravenous administration, and significantly improved SARS-CoV-2-induced pneumonia in a non-human primate model. Furthermore, Tiruthani et al. [60] developed a “muco-trapping” ACE2-Fc decoy with picomolar binding affinity and potent neutralizing capability against both pseudoviruses and live SARS-CoV-2. This decoy demonstrated a significant reduction in viral load, lowering it by up to 10-fold in the nasal turbinates of rodents at 96 h post-infection. However, challenges persist in optimizing lung delivery, particularly concerning the stability of proteins and their potential immunogenicity.

In addition to inhalation therapies, the design of chimeric antigen receptors (CARs) utilizing the miniACE2 protein sequence to target SARS-CoV-2-infected cells is another promising therapeutic strategy. This approach involves genetically modifying natural killer (NK) cells to create an “off-the-shelf” therapy readily available for treating patients with severe COVID-19 (patent pending NC2022/0005322). By leveraging the targeting ability of miniACE2 proteins, CAR-NK cells can be programmed to recognize and eliminate SARS-CoV-2-infected reservoir cells, potentially triggering a robust immune response and improving patient outcomes. Several researchers have validated this strategy [61,62,63,64,65,66]. Another innovative approach involves modifying mesenchymal stromal cells (MSCs) to secrete miniACE2, which could reduce the severe pulmonary effects in critical COVID-19 cases. MSCs with their pulmonary homing signals, can concentrate miniACE2 in the lungs, one of the organs most affected by COVID-19 (patent pending NC2022/0005322). A similar approach was assessed by Wang et al., who engineered MSCs to secrete neutralizing antibodies in vivo to combat SARS-CoV-2 infection [67].

The miniACE2 decoys, BP9 and BP11, computationally designed in this study, exhibited calculated average IC_50_ values of 4.9 and 17.24, respectively, against four SARS-CoV-2 variants, respectively, compared to 107.15 for the wild-type hACE2 (Table 4). These results indicate that BP9 has a 21.9-fold stronger affinity than wild-type hACE2, while BP11 shows a 6.2-fold increase (Table 4). In a prior study, an engineered hACE2 decoy with four mutations (FFWF) demonstrated a roughly 10-fold higher binding affinity for the S protein than wild-type hACE2 [68]. Similarly, Alfaleh et al. constructed IgG1-based WT ACE2-Fc and Modified ACE2-Fc, showing that Modified ACE2-Fc exhibited significantly higher neutralization potency against Omicron SARS-CoV-2 variant compared to WT ACE2-Fc, with up to 16-fold greater inhibition. Our results indicate that the miniACE2 decoy BP9 used in this study potentially surpasses these previous decoys in terms of neutralization efficiency against SARS-CoV-2 variants [69].

Furthermore, using the IC_50_ for hACE2 calculated with the cPass Kit for the BA.1 virus variant, we found that BP9 and BP11 demonstrated greater stability in a human serum matrix compared to the parental hACE2 control. These findings underscore the need for further stability testing of BP9 and BP11 in a preclinical SARS-CoV-2 animal models, particularly via respiratory administration, to validate their potential effectiveness and in vivo stability.

These findings are both promising and significant, as the development of affinity-enhanced tools like miniACE2 decoys is crucial for effectively countering the continuously evolving SARS-CoV-2 variants. MiniACE2 decoys present a compelling alternative, particularly as Omicron subvariants increasingly evade monoclonal antibodies that are initially effective in emergency treatments [70,71]. Moreover, given their demonstrated neutralizing capacity against Omicron subvariants, miniACE2 decoys may also possess the potential to neutralize other *Sarbecoviruses* that utilize ACE2 as an entry receptor, similar to other engineered ACE2 decoys [53,72]. This underscores the potential of miniACE2 decoys as broad-spectrum, pan-coronavirus inhibitors.

Meanwhile, SARS-CoV-2 continues to evolve, with emerging variants such as KP.2 and KP.3, challenging the efficacy of current vaccine designs. This raises concerns that neutralizing titers generated by existing vaccines may not offer sufficiently broad enough protection [73]. As a result, booster vaccinations are necessary to maintain immunity, particularly in immunocompromised individuals, where updated vaccination strategies are crucial for sustaining both humoral and cellular responses [74]. In this context, having pan-coronavirus inhibitors like miniACE2 decoys readily available for administration would be highly advantageous. This approach represents a major step forward in integrating next-generation medical treatments for the global population and provides valuable models for combating future pandemic threats.

Finally, advancing these strategies in low- and middle-income countries could represent a pivotal step in technology transfer and local implementation of next-generation therapies. As demonstrated in this study, computational biology through in silico-guided design offers a robust foundation for developing potential therapeutic agents against SARS-CoV-2. This approach can enhance the independence of local health systems and ultimately improve the health outcomes of vulnerable populations.

## 4. Materials and Methods

### 4.1. In Silico Design of miniACE2 Proteins

Using PDB structures 6LZG [56] and 6M0J [26], the relevant regions for the interaction were located and selected along with the surrounding amino acids (Appendix A). Linker amino acid sequences were added to assemble the discontinuous conformational regions of the ACE2 interface, which interacted with the RBD on the S protein from the SARS-CoV-2 virus.

The fragment sequences, joined as indicated above, were used to model the 3D structures of BPs using the Phyre2 [27] and Robetta servers [28]. The 3D models obtained were visualized using the PyMOL graphical environment [29]. We performed structural alignment for each 3D model using the 6M0J PDB crystal, maintaining the RBD structure in its spatial location. The most promising candidates for Molecular Dynamics (MD) simulations were selected by conducting a preliminary visual evaluation and analyzing the interactions between structures. These proteins were named BP1, BP2, BP3, BP4, BP5, BP6, BP8, BP9, BP10 and BP11 (Appendix A).

### 4.2. Molecular Dynamics Simulation

Once modeled, the coordinates of the proteins selected above in interaction with the RBD were subjected to 100 or 200 ns of unrestricted MD simulation using the Amber18 package (University of California-San Francisco, CA, USA), essentially as previously described [75]. In brief, structures were first solvated with a periodic octahedral pre-equilibrated solvent box using the LEaP module of Amber18, with 12 Å as the shortest distance between any atom in the protein subdomain and periodic box boundaries. Molecular dynamics simulations were conducted using the Particle Mesh Ewald (PME) method for non-bonded interactions with a cutoff distance of 8 Å. The temperature was regulated using a Langevin thermostat, which maintained a fixed temperature of 297 K with a collision frequency of 1 ps. The hydrogen bond constraints were implemented using the SHAKE algorithm, enabling a simulation timestep of 2 fs. To uphold the NPT conditions (constant number of particles, pressure, and temperature), the pressure coupling was managed by a Monte Carlo Barostat set to 297 K and 1 bar. The initial structures underwent 10,000 cycles of energy minimization, followed by a 1 ns restrained equilibration phase, smoothly raising the temperature to 297 K, after which the restraints were gradually removed over 10 ns. Subsequently, each system was subjected to a 100 or 200 ns long free MD production phase. Trajectories were analyzed using cpptraj [76] and VMD [77]. Detailed data are shown in Appendix A.

After the MD procedure, the structures were evaluated using the PRODIGY webserver [30] to obtain the in silico estimated values for the free energy of interaction (ΔG) and the dissociation constant (K_d_). Prior to analysis using PRODIGY, the format of the files containing the coordinates of the protein pairs was modified accordingly. Thus, histidine residues protonated at δ or ε (HID or HIE, in AMBER nomenclature) and cysteine residues forming part of the disulfide bridges (CYX, in AMBER nomenclature) were renamed as HIS or CYS, respectively.

### 4.3. Plasmids Construction, Protein Expression, and Purification

The amino acid sequence of candidate BPs was used to construct the corresponding genes using SnapGene 7.2 Software. The nucleotide sequence was optimized for expression in *E. coli* and restriction targets to facilitate cloning were included. The plasmids pEcoBP9 and pEcoBP11 were synthesized by Gene Universal (Newark, DE, USA) (Appendix A) and used separately to transform the BL21(DE3) *E. coli* strain. Protein expression was assessed by Gene Universal (Newark, DE, USA). Briefly, protein expression was induced with 0.5 mM IPTG at 37 °C. Then, the solubility was analyzed by IPTG induction at a final concentration of 0.2 mM or 1.0 mM at 37 °C or 15 °C by SDS-PAGE. After ultrasonication (500 W of power for 6 min, with an interval of 6 s every 3 s of operation) and centrifugation at 8.000 g/4 °C/5 min, the supernatant was recovered, and the pellet was lysed with PBS buffer, 8 M Urea, pH 7.4. The retrieved target proteins in the pellet were purified by affinity with Ni-NTA beads (Profinity™ IMAC Resin, Ni-charged, Bio-Rad) and IEX purification (Q Sepharose Fast Flow, Cytiva). The final elution samples were pooled and dialyzed in PBS buffer, 300 mM NaCl, 2 mM DTT, pH 7.4, and analyzed by western blot (Appendix A). The ACE2 protein encoded in the plasmid pcDNA-ACE2-His (synthesized for Gene Universal, Newark, DE, USA) was expressed using the Expi293 Expression System in accordance with the manufacturer’s recommendations and purified by affinity with Ni-NTA beads (Profinity™ IMAC Resin, Ni-charged, Bio-Rad) (Appendix A).

### 4.4. SARS-CoV-2 ELISA Neutralization Assay

Proteins obtained from *E. coli* were quantified using BSA titration on SDS-PAGE gels, which were subsequently stained with Coomassie Blue. Gel images were acquired and quantified using the ImageJ 1.8.0 software. The protein concentration was adjusted to perform ELISA using the commercially available cPass SARS-CoV-2 Neutralization Antibody Detection Kit (Nanjing GenScript Diagnostics Technology Co, Jiangsu Province China Inc.).

The ELISA test was verified under environmental conditions using equipment available at the IDCBIS infrastructure. A 96-well plate coated with the hACE2 protein was tested with various concentrations of each BP in the presence of different RBD proteins (Wuhan, Mu, Omicron BA.1, and Omicron BA. 2 variants/subvariants). To evaluate the amount of RBD bound to hACE2, the kit utilized RBD conjugated with a peroxidase enzyme, resulting in the degradation of 3,3′,5,5′-tetramethylbenzidine (TMB) substrate proportional to the RBD binding to ACE2 at the bottom of the 96-well plate. The intensity of the resulting color reaction was inversely proportional to the RBD-neutralization activity of the blocking protein added to the wells. All experiments were conducted following the manufacturer’s instructions. Briefly, proteins were diluted in sample dilution buffer and incubated with RBD-HRP solution for 30 min at 37 °C. Subsequently, the samples were plated in in triplicate in a 96-well plate included in the kit and incubated at 37 °C for 15 min. The plates were then washed and incubated with TMB solution, protected from light, at room temperature (18–25 °C) for 25 min. Finally, a stop solution was added and the absorbance was immediately read at 450 nm. Each run was validated according to predefined criteria provided by the kit.

### 4.5. Statistics and Software Tools

Protein modeling was conducted using the Protein Homology/analogY Recognition Engine V 2.0 (Phyre2.0: http://www.sbg.bio.ic.ac.uk/phyre2/html/page.cgi?id=index 10 September 2024) and the Robetta protein structure prediction service (https://robetta.bakerlab.org/ 31 August 2024). Molecular dynamics simulations were performed with Amber v.18 and AmberTools v.18 (University of California, San Francisco, CA, USA). Analysis of the molecular dynamics trajectories was carried out using Visual Molecular Dynamics (VMD) version 1.9.3 (University of Illinois Urbana-Champaign, IL, USA). The In silico assessment of ΔG and K_d_ values was performed using the PRODIGY web server (Utrecht Biomolecular Interaction Web, Utrecht University, Utrecht, The Netherlands).

All the BPs concentrations were run in triplicates, and the results were calculated following the manufacturer’s instructions. The results obtained were analyzed by GraphPad Prism v. 9.0.0. For IC_50_, dilutions of BPs were prepared in a dilution buffer. IC_50_ values were derived from the log(inhibitor) vs. response variable slope (four parameters). The significance of the IC_50_ values obtained for BPs in comparison to the performance of hACE2 was assessed using an ANOVA test followed by Tukey’s multiple comparison test. The significance of the stability test was determined using one-way ANOVA.

## 5. Patents

**Patent application File No**. NC2022/0005322. **Title**: Proteínas miniACE2 solubles que interaccionan con SARS-CoV-2 y usos de las mismas. **Type of Procedure**: National Invention Patent. **Applicant**: CONSEJO SUPERIOR DE INVESTIGACIONES CIENTÍFICAS (CSIC), INSTITUTO DISTRITAL DE CIENCIA BIOTECNOLOGIA E INNOVACIÓN EN SALUD—IDCBIS. **Filing Date**: 27 April 2022. **Inventors**: Paulino Gómez-Puertas, Cesar Augusto Ramirez Segura.

## Figures and Tables

**Figure 1 ijms-25-10802-f001:**
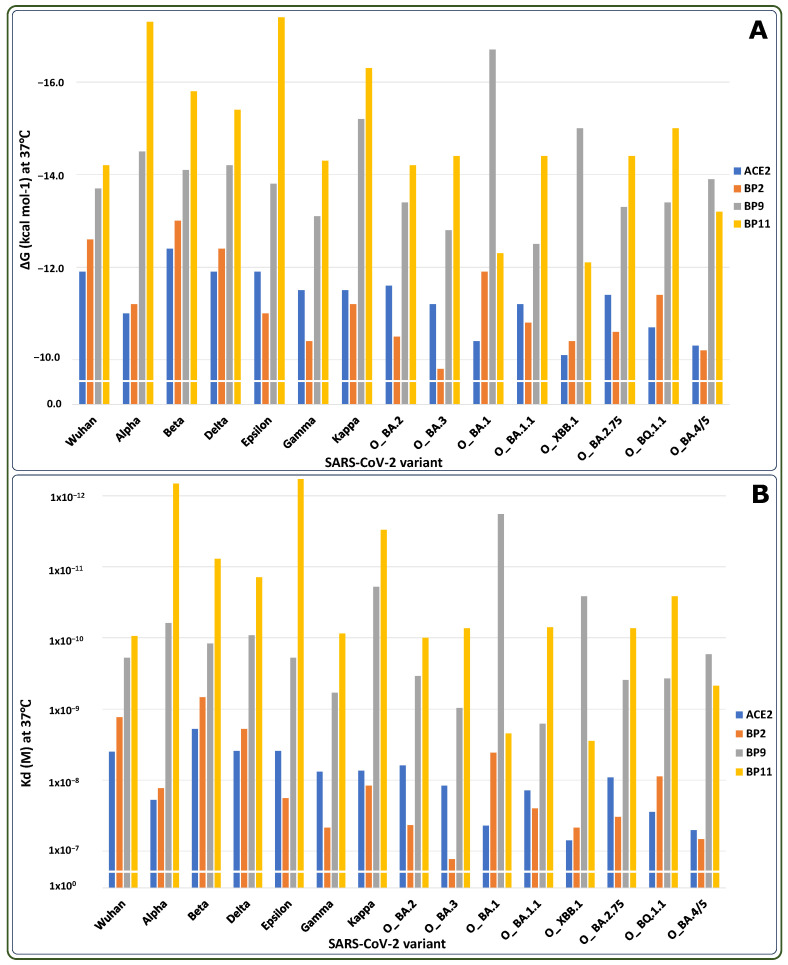
After 200 ns of MD simulation, the ΔG (**A**) and K_d_ (**B**) values were calculated using the PRODIGY web server [30] for the interaction of each BP with a broad range of RBDs from SARS-CoV-2 viral variants: (Wuhan, PDB: 6m0j [26]; Alpha B.1.1.7, PDB: 8DLK [31]; Beta, PDB: 7VX4 [32]; Delta, PDB: 7W9I [33]; Epsilon, PDB: 8DLV [31]; Gamma, PDB: 8DLQ [31]; Kappa, PDB: 7V86 [34]; Omicron BA.2, PDB: 7ZF7 [35]; Omicron BA.3, PDB: 7XB1 [36]; Omicron BA.1, PDB: 7U0N [37]; Omicron BA.1.1, PDB: 7XAZ [36]; Omicron XBB.1, PDB: 8IOV [38]; Omicron BA.2.75, PDB: 8ASY [39]; Omicron BQ.1.1, PDB: 8IF2 [40]; Omicron BA.4/5, PDB: 8AQS [41]).

**Figure 2 ijms-25-10802-f002:**
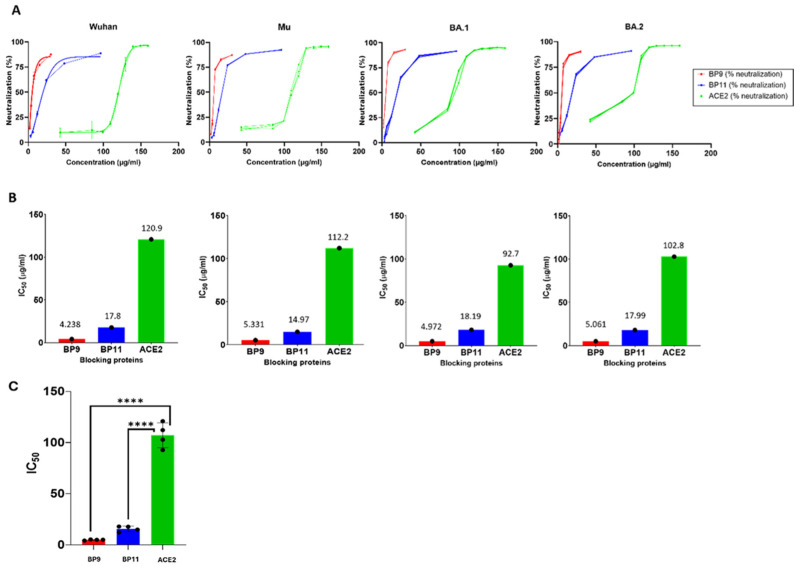
In silico designs of BPs expressed in *E. coli* broadly neutralize SARS-CoV-2 variants through ELISA assays. (**A**) Neutralization efficacy of BP9 and BP11 ACE2 decoys against Wuhan (wild type), Mu, BA.1, and BA.2 variants compared to ACE2 protein, showing enhanced efficacy and potential for novel BP designs (*n* = 3 technical replicates shown). (**B**) IC_50_ comparative panel between novel BPs and ACE2, suggesting their potential therapeutic use at low doses. (**C**) Statistical analyses (Tukey’s multiple comparison test performed using GraphPad Prism V10) were performed to calculate the statistical significance of the BPs (BP9 and BP11) compared with the parental hACE2. **** Represent the level of significance.

**Figure 3 ijms-25-10802-f003:**
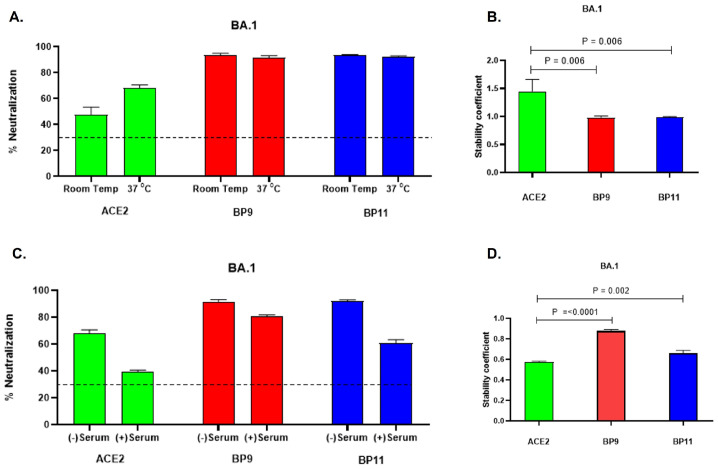
Stability of BP9 and BP11 at 37 °C and the Serum Matrix. Stability was evaluated by measuring neutralization activity using the cPass SARS-CoV-2 Neutralization Antibody Detection Kit for the BA.1 virus variant, with the ACE2 IC_50_ value (92.92 µg/mL) as a standard protein concentration under the effects of temperature and serum matrix. (**A**): Proteins were incubated at room temperature and at 37 °C for 2 h to assess their stability. (**C**) Proteins were incubated at 37 °C for 2 h in the presence or absence of human serum samples, which were confirmed to lack neutralizing antibodies against SARS-CoV-2 (Appendix A). Each experiment was performed in three technical replicates. (**B**,**D**): The stability coefficient was calculated to identify significant differences in the stability of BP9 and BP11 compared to that of hACE2. The dotted line represents the neutralization cutoff as specified by the kit’s manufacturer. Data are presented as the mean ± standard deviation (SD). *p-*values were determined using one-way ANOVA (**B**,**D**).

**Table 1 ijms-25-10802-t001:** Interactions of BPs with the RBD measured at 37.0 °C by the PRODIGY web server.

Protein Name	Number of aa	100 ns RMSD Value	Interactions of BPs with the RBD Measured at 37.0 °C by PRODIGY Web Server											
			ΔG (kcal mol^−1^)	K_d_ (M)	K_d_ hACE2/K_d_ BP	Number of Interfacial Contacts (ICs) per Property						Non-Interacting Surface (NIS) per Property		
						Charged-Charged	Charged-Polar	Charged-Apolar	Polar-Polar	Polar-Apolar	Apolar-Apolar	Charged	Apolar	Total
BP1	137	5.717	−12.5	1.4 × 10^−9^	2.86	5	13	26	10	23	11	21.03	36.53	57.56
BP2	118	4.3356	−12.6	1.3 × 10^−9^	3.08	3	12	20	3	20	23	19.31	37.07	56.38
BP3	147	5.0111	−11.7	5.3 × 10^−9^	0.75	5	11	20	5	19	9	21.31	37.8	59.11
BP4	132	5.6667	−13.6	2.5 × 10^−10^	16.00	2	13	17	4	28	20	19.26	38.15	57.41
BP5	135	5.9973	−10.6	3.4 × 10^−8^	0.12	1	11	13	0	12	7	20.51	35.53	56.04
BP6	124	4.4747	−10.8	2.6 × 10^−8^	0.15	1	12	15	2	16	13	19.7	39.02	58.72
BP8	118	3.8467	−12	3.6 × 10^−9^	1.11	3	13	21	2	16	12	19.61	36.86	56.47
BP9	132	3.2181	−13.7	2.1 × 10^−10^	19.05	0	14	19	6	30	20	19.71	37.59	57.3
BP10	118	4.5989	−9.6	1.7 × 10^−7^	0.02	2	10	10	1	10	15	19.47	37.02	56.49
BP11	132	4.1444	−14.2	1.0 × 10^−10^	40.00	5	10	24	4	27	23	18.98	39.42	58.4
ACE2 PDB: 6m0j	768	----	−11.9	4.0 × 10^−9^	1.00	3	10	18	4	21	10	25.73	35.06	60.79

**Table 2 ijms-25-10802-t002:** ΔG values for a broad range of SARS-CoV-2 viral variants (columns ↓). After 200 ns of MD simulations, the ΔG values were calculated using the PRODIGY web server [30] and recorded for each Blocking Protein (rows →), ACE2, BP2, BP9, and BP11 with the fifteen SARS-CoV-2 variants listed below.

Blocking Protein → RBD-Variant ↓	ΔG (kcal mol^−1^) at 37 °C			
	hACE2	BP2	BP9	BP11
Wuhan	−11.9	−12.6	−13.7	−14.2
Alpha	−11	−11.2	−14.5	−17.3
Beta	−12.4	−1	−14.1	−15.8
Delta	−11.9	−12.4	−14.2	−15.4
Epsilon	−11.9	−11	−13.8	−17.4
Gamma	−11.5	−10.4	−13.1	−14.3
Kappa	−11.5	−11.2	−15.2	−16.3
O_BA.2	−11.6	−10.5	−13.4	−14.2
O_BA.3	−11.2	−9.8	−12.8	−14.4
O_BA.1	−10.4	−11.9	−16.7	−12.3
O_BA.1.1	−11.2	−10.8	−12.5	−14.4
O_XBB.1	−10.1	−10.4	−15	−12.1
O_BA.2.75	−11.4	−10.6	−13.3	−14.4
O_BQ.1.1	−10.7	−11.4	−13.4	−15
O_BA.4/5	−10.3	−10.2	−13.9	−13.2
**ΔG** $\bar{x}$**→**	−11.27	−11.16	−13.97	−14.71

**Table 3 ijms-25-10802-t003:** K_d_ values for a broad range of SARS-CoV-2 viral variants (columns ↓). After 200 ns of MD simulations, the K_d_ values were calculated using the PRODIGY web server [30] and recorded for each Blocking Protein (rows →), hACE2, BP2, BP9, and BP11 with the fifteen SARS-CoV-2 variants listed below.

Blocking Protein → RBD-Variant ↓	K_d_ at 37 °C			
	ACE2	BP2	BP9	BP11
Wuhan	4.00 × 10^−9^	1.30 × 10^−9^	1.90 × 10^−10^	9.40 × 10^−11^
Alpha	1.90 × 10^−8^	1.30 × 10^−8^	6.20 × 10^−11^	6.70 × 10^−13^
Beta	1.90 × 10^−9^	6.80 × 10^−10^	1.20 × 10^−10^	7.70 × 10^−12^
Delta	3.90 × 10^−9^	1.90 × 10^−9^	9.20 × 10^−11^	1.40 × 10^−11^
Epsilon	3.90 × 10^−9^	1.80 × 10^−8^	1.90 × 10^−10^	5.80 × 10^−13^
Gamma	7.60 × 10^−9^	4.70 × 10^−8^	5.90 × 10^−10^	8.70 × 10^−11^
Kappa	7.40 × 10^−9^	1.20 × 10^−8^	1.90 × 10^−11^	3.00 × 10^−12^
O_BA.2	6.20 × 10^−9^	4.30 × 10^−8^	3.40 × 10^−10^	1.00 × 10^−10^
O_BA.3	1.20 × 10^−8^	1.30 × 10^−7^	9.60 × 10^−10^	7.30 × 10^−11^
O_BA.1	4.40 × 10^−8^	4.10 × 10^−9^	1.80 × 10^−12^	2.20 × 10^−9^
O_BA.1.1	1.40 × 10^−8^	2.50 × 10^−8^	1.60 × 10^−9^	7.10 × 10^−11^
O_XBB.1	7.10 × 10^−8^	4.70 × 10^−8^	2.60 × 10^−11^	2.80 × 10^−9^
O_BA.2.75	9.20 × 10^−9^	3.30 × 10^−8^	3.90 × 10^−10^	7.30 × 10^−11^
O_BQ.1.1	2.80 × 10^−8^	8.90 × 10^−9^	3.70 × 10^−10^	2.60 × 10^−11^
O_BA.4/5	5.10 × 10^−8^	6.80 × 10^−8^	1.70 × 10^−10^	4.70 × 10^−10^
K_d_ $\bar{x}$ →	1.89 × 10^−8^	3.02 × 10^−8^	3.41 × 10^−10^	4.01 × 10^−10^
K_d_ ACE2 $\bar{x}$/K_d_ BP $\bar{x}$	1.0	0.63	55.28	47.03

**Table 4 ijms-25-10802-t004:** Recorded IC_50_ values for ACE2, BP2, BP9, and BP11 in interactions with each of the SARS-CoV-2 variants.

SARS-CoV-2 Variant BPs	Experimentally Assessed IC_50_ Values				IC_50_ $\bar{x}$	$\bar{x}$ ACE2 IC_50_/$\bar{x}$ BPs IC_50_
	Wuhan	Mu	O_BA.1	O_BA.2		
**BP9**	**4.24**	5.33	4.97	5.06	**4.90**	**21.87**
**BP11**	17.8	14.97	18.19	17.99	**17.24**	**6.22**
**ACE2**	120.9	112.2	92.7	102.8	**107.15**	**1.00**

**Table 5 ijms-25-10802-t005:** Percentage of neutralization of BP9, BP11, and hACE2 under three different conditions: room temperature, 37 °C, and influence of the human serum matrix.

Protein	% Neutralization											
	Temperature								Serum			
	Room Temperature			$\bar{X}$	37 °C			$\bar{X}$	S1	S2	S3	$\bar{X}$
	R1	R2	R3		R1	R2	R3					
**ACE2**	**48.81**	**52.85**	**41.42**	**47.69**	**70.12**	65.66	69.09	68.29	40.57	38.18	39.21	39.32
BP9	93.75	94.7	91.96	93.47	90.28	91.88	92.95	91.7	79.42	81.82	80.35	80.53
BP11	93.41	93.41	93.94	93.58	92.61	92.72	91.39	92.24	58.26	62.65	61.72	60.88

## Data Availability

Additional supporting data are available from the corresponding authors upon reasonable request.

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
