# Peer review of "In Silico Design of miniACE2 Decoys with In Vitro Enhanced Neutralization Activity against SARS-CoV-2, Encompassing Omicron Subvariants"

_ijms, 2024, doi:10.3390/ijms251910802_

Round 1
Reviewer 1 Report
Comments and Suggestions for Authors
The manuscript titled “In-silico design of miniACE2 decoys with in-vitro enhanced neutralization activity against SARS-CoV-2, encompassing Omicron Subvariants” presents a compelling investigation into the potential of miniACE2 decoys as a novel therapeutic strategy. The study demonstrates that blocking proteins (BPs) exhibit strong binding affinities to various SARS-CoV-2 variants, with BP9 showing an average IC50 of 4.9 μg/ml, highlighting its efficacy as a treatment against evolving strains of the virus. Given the significance of this research topic, it warrants acceptance following the resolution of several concerns.
1. The manuscript inconsistently uses commas instead of decimal points in numerical data, which may lead to confusion. It is recommended to standardize the use of decimal points throughout the manuscript for clarity.
2. The Y-axis label is missing in Figure 2A and should be added to enhance interpretability.
3. It is essential to include relevant references that discuss the neutralization capacity of similar therapeutic strategies to provide context and support for the findings presented.
4. The sentence on page 9, lines 240-241, should be revised for clarity. Modify “Here, utilizing the PDB structures 6LZG the PDB structures 6LZG[56] and 6M0J[29], which were available at the onset of the pandemic and three miniACE2 proteins were designed” with “Here, utilizing the PDB structures 6LZG and 6M0J, which were available at the onset of the pandemic, three miniACE2 proteins were designed.”
5. Lines 88 to 100 in the Results section contain computational details that may not be necessary for the reader and could be removed to streamline the presentation.
6. To improve visual clarity, it is suggested to remove the black background from Figure 1.
7. Clarification is needed regarding the selection process for BP2, BP9, and BP11 out of the initial set (BP1, BP2, BP3, BP4, BP5, BP6, BP8, BP9, BP10, and BP11) for further studies.
8. The manuscript should specify how many amino acids are present in the designed miniACE2 decoys to provide detailed insight into their structural characteristics.
9. Mention possible limitation of experimental studies specifically toxicity related concerns.
Comments on the Quality of English LanguageMinor English editing is needed to correct typographical and grammatical errors.
Author Response
Comment 1. The manuscript inconsistently uses commas instead of decimal points in numerical data, which may lead to confusion. It is recommended to standardize the use of decimal points throughout the manuscript for clarity.
Response: Thank you for your feedback. We have made the requested changes and standardized the use of decimal points throughout the manuscript.
Comment 2. The Y-axis label is missing in Figure 2A and should be added to enhance interpretability.
Response: Thank you for your observation. The addition of the Y-axis label was successfully added to improve the figure interpretation.
Comment 3. It is essential to include relevant references that discuss the neutralization capacity of similar therapeutic strategies to provide context and support for the findings presented.
Response: Thank you for your suggestion. We have included relevant references discussing the neutralization capacity of similar therapeutic strategies in the revised manuscript.
The miniACE2 decoys, BP9 and BP11, computationally designed in this study, exhibited calculated average IC50 values of 4.9 and 17.24, respectively against four SARS-CoV-2 variants, respectively, compared to 107.15 for the wild-type hACE2 (Table 4). These results indicate that BP9 has a 21.9-fold stronger affinity than wild-type hACE2, while BP11 shows a 6.2-fold increase (Table 4). In a prior study, an engineered hACE2 decoy with four mutations (FFWF) demonstrated a roughly 10-fold higher binding affinity to the S protein than wild-type hACE2 [68]. Similarly, Alfaleh et al constructed IgG1-based WT ACE2-Fc and Modified ACE2-Fc, showing that ModifiedACE2-Fc exhibited significantly higher neutralization potency against Omicron SARS-CoV-2 variant compared to WT ACE2-Fc, with up to 16-fold greater inhibition. Our results indicate that the miniACE2 decoy BP9, used in this study, potentially surpasses these previous decoys in terms of neutralization efficiency against SARS-CoV-2 variants.[69].
Comment 4. The sentence on page 9, lines 240-241, should be revised for clarity. Modify “Here, utilizing the PDB structures 6LZG the PDB structures 6LZG[56] and 6M0J[29], which were available at the onset of the pandemic and three miniACE2 proteins were designed” with “Here, utilizing the PDB structures 6LZG and 6M0J, which were available at the onset of the pandemic, three miniACE2 proteins were designed.
Response: Thank you for your kind appreciation. We have revised the sentence to incorporate your suggestion in the revised manuscript.
Pathophysiological studies of SARS-CoV-2 infection have highlighted the potential utility of recombinant hACE2 proteins, demonstrating their effectiveness in neutralizing a wide range of SARS-CoV-2 variants In-vitro. So, ACE2 decoys are emerging as promising candidates for emergency treatment [53], [54], [55]. Here, utilizing the PDB structures 6LZG[56] and 6M0J[26], which were available at the onset of the pandemic, three miniACE2 proteins were designed. BP2 was created to include the RBD-interacting surface, incorporating the glycosylated asparagine residues N53 and N90 from the ACE2 sequence (corresponding to N33 and N70 on BP2).
Comment 5. Lines 88 to 100 in the Results section contain computational details that may not be necessary for the reader and could be removed to streamline the presentation.
Response: Thank you for your feedback. We understand your concern regarding the computational details in lines 88 to 100 of the Results section. To improve the clarity and flow of the manuscript, we rewrite this paragraph to include essential information about the protein design as follow:
Through the analysis of the RBD/hACE2 interaction, based on the PDB: 6M0J structure [26], key regions of ACE2, specifically amino acids S19–S106 and Q340–A386, corresponding to the interaction surface, were identified for incorporation into the miniACE2 design. These discontinuous conformational regions of the ACE2 protein were linked in various ways to reconstruct the RBD-interacting surface, while excluding both the catalytic domain and non-interacting regions of ACE2.
We appreciate your suggestion, as it will help make the presentation more concise and accessible for the reader.
Comment 6. To improve visual clarity, it is suggested to remove the black background from Figure 1.
Response: Thank you for the suggestion. We have removed the black background from Figure 1 to improve visual clarity in the revised manuscript.
Comment 7. Clarification is needed regarding the selection process for BP2, BP9, and BP11 out of the initial set (BP1, BP2, BP3, BP4, BP5, BP6, BP8, BP9, BP10, and BP11) for further studies.
Response: Thank you for your feedback. We have added clarification in the revised manuscript regarding the selection criteria.
The proteins selected above in interaction with the RBD were subjected to 100 ns of unrestricted MD simulation (Supplementary Table 1 and Supplementary Figure 1). The files obtained as results of the MD simulations were evaluated using the PRODIGY webserver[30], to obtain the In-silico estimated values for ΔG and Kd. To assess the potential increased capacity to interact with the RBD compared to the natural ACE2 receptor, the ACE2 Kd / BP Kd ratio was calculated for each BP (Table 1), then, the top 5 BPs according to the highest ACE2 Kd / BP Kd ratios were selected (BP11, BP9, BP4, BP2 and BP1), (Table 1). Out of these 5 BPs, those with the most stable structures over the 100 ns MD simulation, as indicated by the lowest RMSD values (Table 1), were selected to continue the In-silico study. Consequently, three proteins exhibiting the highest ACE2 Kd / BP Kd ratios and the most stable structures (BP9, BP2, and BP11) were chosen as candidate BPs,
Comment 8. The manuscript should specify how many amino acids are present in the designed miniACE2 decoys to provide detailed insight into their structural characteristics.
Response: Thank you for your observation. While the specification of the number of amino acids is provided in Table 1, we have included the requested information in line 115, for improved clarity.
Consequently, three proteins exhibiting the highest ACE2 Kd / BP Kd ratios and the most stable structures (BP9, BP2, and BP11) were chosen as candidate BPs, with lengths ranging from 118 to 132 amino acids (Supplementary Figure 2, Table 1)
Comment 9. Mention possible limitation of experimental studies specifically toxicity related concerns.
Response: Thank you for your valuable suggestion. We have addressed the potential limitations of the experimental studies, with a particular focus on toxicity concerns, in the revised manuscript by including the following paragraph.
Although no adverse effects are anticipated from the natural activity of ACE2, given that the catalytic site is absent in miniACE2, it remains essential to evaluate potential immune responses or interactions with other human proteins that could lead to toxicity. This is particularly important since linker amino acid sequences were incorporated into the miniACE2 design, which may introduce unforeseen immunogenicity or off-target effects. Therefore, thorough preclinical testing will be crucial to assess safety.
Comment 10. Comments on the Quality of English Language:
Minor English editing is needed to correct typographical and grammatical errors.
Response: Thank you for the comment. Minor typographical and grammatical errors have been corrected in the revised manuscript.
Reviewer 2 Report
Comments and Suggestions for Authors
I have reviewed the manuscript submitted by Jenny Andrea Arévalo-Romero et al. and am pleased to recommend its acceptance for publication in IJMS. The manuscript presents a cutting-edge study on the design and development of miniACE2 decoys as a novel therapeutic strategy against SARS-CoV-2 and its variants, addressing a critical need for effective treatments.
The authors employ a robust methodology, combining in-silico design, molecular dynamics, and in-vitro testing. Their findings demonstrate promising neutralizing effects, particularly with miniACE2 BP9, which exhibited potent neutralizing capacity against multiple variants. The manuscript is well-written, clearly organized, and provides significant contributions to the field of COVID-19 research and therapeutic development.Overall, I rate this manuscript as excellent and strongly recommend its acceptance. The study's innovative approach, rigorous methodology, and significant potential for translational medicine make it an invaluable contribution to IJMS.
Thank you for considering my evaluation.
Comments on the Quality of English Language
NA
Author Response
Thank you for your positive and thorough review of our manuscript. We are grateful for your support and are pleased that you found our study on miniACE2 decoys both innovative and valuable for the ongoing efforts in COVID-19 therapeutic development. Your comments on the robustness of our methodology and the potential translational impact of our findings, particularly regarding miniACE2 BP9, are greatly appreciated. We are excited to contribute to this field and look forward to the publication process.
Reviewer 3 Report
Comments and Suggestions for Authors
In this article, the authors employed an in-silico approach to design blocking proteins (BPs) with strong binding affinity to the receptor binding domain of multiple SARS-CoV-2 variants and human ACE2. The BPs were also tested in vitro for their neutralizing effects. Based on both in-silico and in-vitro results, the authors identified BP9 as a promising therapeutic candidate for combating SARS-CoV-2 and its variants. However, the authors should provide a more detailed discussion of the results in the Discussion and Conclusion sections. Overall, this manuscript presents a compelling combination of in-silico design, molecular modeling, molecular dynamics simulations, and in-vitro data to characterize BPs targeting the receptor binding domain of SARS-CoV-2.
Author Response
Thank you for your thoughtful review and valuable feedback. We appreciate your recognition of the combination of in-silico design and in-vitro data in our study. We have revised the discussion and conclusion sections to provide a more detailed analysis of the results, as suggested, ensuring a clearer interpretation of the findings and their broader implications. Your insights will help strengthen the manuscript, and we are grateful for your recommendations.